# Piceatannol as an Antiviral Inhibitor of PRV Infection In Vitro and In Vivo

**DOI:** 10.3390/ani13142376

**Published:** 2023-07-21

**Authors:** Zhiying Wang, Xiaojing Cai, Zhiyuan Ren, Yi Shao, Yongkang Xu, Lian Fu, Yan Zhu

**Affiliations:** College of Veterinary Medicine, Northeast Agricultural University, Harbin 150038, China; wzyzy0419@163.com (Z.W.); caixiaojing777@126.com (X.C.); zhiyuanren0328@163.com (Z.R.); sy15049522319@163.com (Y.S.); 18754963560@163.com (Y.X.); a1473704626@163.com (L.F.)

**Keywords:** piceatannol, pseudorabies virus, antiviral activity

## Abstract

**Simple Summary:**

PRV can infect most mammals, mainly affecting the nervous and reproductive systems of infected animals, and can cause the death of piglets and other susceptible animals. Identifying effective antiviral agents against PRV to prevent a latent infection is essential. Piceatannol is a bioactive polyphenol substance. It is mainly found in grapes, mushrooms, blueberries, passion fruit and other edible fruits and vegetables. In this study, we evaluated the inhibitory effect of piceatannol on PRV replication in vivo and in vitro and investigated the mechanism of action of piceatannol against PRV. Piceatannol could exert an anti-PRV effect by reducing the transcription level of viral genes, reducing PRV-induced apoptosis and elevating the levels of IL-4, TNF-α and IFN-γ in the serum of mice. Animal experiments showed that piceatannol could delay the onset of disease, reduce the viral load in the brain and kidney and reduce the pathological changes in the tissues and organs of the mice to improve the survival rate of the mice (14.3%). Therefore, the anti-PRV activity of piceatannol in vivo and in vitro was systematically evaluated in this study to provide scientific data for developing a new alternative measure for controlling PRV infection.

**Abstract:**

Pseudorabies virus (PRV) belongs to the family Herpesviridae. PRV has a wide host range and can cause cytopathic effects (CPEs) in PK-15 cells. Therefore, PRV was used as a model to study the antiviral activity of piceatannol. The results showed that piceatannol could restrain PRV multiplication in PK-15 cells in a dose-dependent manner. The 50% inhibitory concentration (IC_50_) was 0.0307 mg/mL, and the selectivity index (SI, CC_50_/IC_50_) was 3.68. Piceatannol could exert an anti-PRV effect by reducing the transcription level of viral genes, inhibiting PRV-induced apoptosis and elevating the levels of IL-4, TNF-α and IFN-γ in the serum of mice. Animal experiments showed that piceatannol could delay the onset of disease, reduce the viral load in the brain and kidney and reduce the pathological changes in the tissues and organs of the mice to improve the survival rate of the mice (14.3%). Therefore, the anti-PRV activity of piceatannol in vivo and in vitro was systematically evaluated in this study to provide scientific data for developing a new alternative measure for controlling PRV infection.

## 1. Introduction

Pseudorabies virus (PRV) is a double-stranded DNA virus belonging to the family Herpesviridae, a member of the genus Varicellovirus, and the causative agent of pseudorabies (also known as Aujeszky disease, AD) furthermore wild swine may be the source of PRV infection [1]. PRV can infect most mammals, mainly affecting the nervous and reproductive systems of infected animals, and can cause the death of piglets and other susceptible animals. In addition, PRV induces a cytopathic effect (CPE) in infected cells [2]. Therefore, PRV is an important model virus for studying the herpesvirus’s biology and effects on neuronal pathways. Recent studies have shown that PRV can infect humans and cause central nervous system diseases, endophthalmitis and encephalitis [3,4].

Natural compounds have the advantages of being available from a wide range of sources, exhibiting low toxicity and having a wide range of biological activities [5]. Piceatannol is a bioactive polyphenol substance. It is mainly found in grapes, mushrooms, blueberries, passion fruit and other edible fruits and vegetables [6,7,8]. It has been proven to have anti-inflammatory, antioxidation, antibacterial, antiaging, antileukemia, anti-cell proliferation, immunomodulatory, cardiovascular-protective and anticancer traits in addition to offering other biological activities [9,10,11] continuously. In addition, it has been shown that polyphenols can inhibit human and animal viruses. Piceatannol has an inhibitory effect on human cytomegalovirus in vitro [12]. Polyphenol stilbene compounds can inhibit HIV-1 integrase in vitro and, thus, inhibit HIV-1 proliferation [13]. In this study, we evaluated the inhibitory effect of piceatannol on PRV replication in vivo and in vitro and investigated the mechanism of action of piceatannol against PRV, aiming to provide some new ideas for the control of PRV infection.

## 2. Materials and Methods

### 2.1. Cells, Materials and Piceatannol

PK-15 cells were cultured in DMEM supplemented with 10% fetal bovine serum. The medium used for cytotoxicity and antiviral tests contained 2% fetal bovine serum. PRV-TJ (GenBank accession: KJ789182.1) was provided by Harbin Veterinary Institute of China. PRV was subpropagated in PK-15 cells, and the TCID_50_ was calculated with the Reed-Muench method to be 106.54/0.1 mL. The stock was stored at −80 °C until use. Piceatannol was purchased from Dalian Meilun Biotechnology Co., Ltd. (Dalian, China), with a purity of 98%, and was dissolved in 1% dimethyl sulfoxide (DMSO).

### 2.2. Determination of Cytotoxicity and Inhibitory Activity

Cell viability was evaluated using a CCK-8 kit (Bimake, Houston, TX, USA) to assess the cytotoxicity of piceatannol. In short, PK-15 cells in 96-well plates were treated with a serial twofold dilution of piceatannol and incubated at 37 °C and 5% CO_2_ for 48 h. At the same time, a group of cells without continuous s was maintained as a control. After 48 h, CCK-8 solution was added according to the instructions, and the cultures were incubated at 37 °C for 45 min. The OD value of each well at 450 nm was measured with a plate reader, and the cell survival rate was calculated to evaluate the cytotoxicity of piceatannol. To determine the inhibitory activity of piceatannol against PRV, different concentrations of piceatannol were mixed with an equal volume of 100 TCID_50_ of PRV solution at a nontoxic concentration and incubated at 37 °C for 1 h. The above mixture was added to 96-well cell culture plates, which were incubated for 1 h and then washed with PBS, and cell maintenance solution was then added. Forty-eight hours later, the OD values at 450 nm of each well were determined, and the virus inhibition rate was calculated to evaluate the inhibitory activity of piceatannol against PRV. The half inhibitory concentration (IC_50_) and half cytotoxic concentration (CC_50_) of piceatannol were calculated with GraphPad Prism 8.0 software.

### 2.3. Effect of Piceatannol on the PRV Life Cycle

Virus adsorption phase: Six-well plates were inoculated with PK15 cells at a density of 10^5^ cells/well. After cooling the cells at 4 °C for 30 min, piceatannol (0.063, 0.032 mg/mL) and virus (100 TCID_50_) premix were added and adsorbed at 4 °C for 1 h, and then the cells were washed to remove unadsorbed viruses, followed by incubation at 37 °C.

Virus entry phase: Six-well plates were inoculated with PK15 cells at a density of 10^5^ cells/well. The PK15 cell monolayer was incubated with the virus (100 TCID_50_) at 4 °C for 1 h, the virus solution was removed and fresh medium containing piceatannol (0.063, 0.032 mg/mL) was added.

Virus replication phase: Six-well plates were inoculated with PK15 cells at a density of 10^5^ cells/well. The PK15 cell monolayer was incubated with the virus (100 TCID_50_) at 37 °C for 1 h, the virus solution was removed and fresh medium containing piceatannol (0.063, 0.032 mg/mL) was added.

Infected cells were harvested 48 h after infection. A TIANamp Genomic DNA Kit (Tiangen Biotech; Beijing, China) was used to extract the viral genomic DNA, and FQ-PCR was used to analyze the viral DNA copy number in infected cells. The reaction primers were 5′-GCCGAGTACGACCTCTGCC-3′ (forward) and 5′-CGAGACGAACAGCAGCCG-3′ (reverse), and the probe was 5′-HEX-CCGCGTGCACCACGAAGCCT-BHQ1. A standard plasmid containing the gI gene was used to generate a standard curve, and the FQ-PCR conditions were predenaturation at 94 °C for 10 min, followed by 45 cycles of 94 °C for 35 s and 60 °C for 35 s.

### 2.4. Effects of Piceatannol on PRV Gene Expression

The PK15 cells that grew as the monolayer were added to the mixture of piceatannol and PRV (100TCID_50_) for 1 h. Then, the supernatant was removed and replaced with fresh medium. Samples of infected cells were collected at 3, 6, 12, 24 and 48 h after viral infection. Total RNA was extracted with an RNA Easy Fast Tissue/Cell Kit (Tiangen Biotech; Beijing, China). According to the instructions for FastKing gDNA Dispelling RT SuperMix, reverse transcription was carried out, and qPCR primers were designed using Primer 5 software according to the PRV gene and β-actin gene sequence. The RT-qPCR procedure was as follows: predenaturation at 95 °C for 5 min, followed by 95 °C for 15 s, 55 °C for 30 s and 72 °C for 35 s for a total of 40 cycles. The primer sequences are shown in Table 1.

### 2.5. Apoptosis Analysis

The samples were analyzed with an Annexin V-FITC apoptosis kit (Dalian Boglin Biotechnology Co., Ltd., Dalian, China), and cell apoptosis was assessed with flow cytometry. In brief, monolayers of PK-15 cells were washed twice with PBS, and a mixture of 100 TCID_50_ PRV virus solution and an equal volume of 0.063 mg/mL piceatannol that had been treated at 37 °C for 1 h was added. After 1 h, the solution was discarded, the cells were washed with PBS and then the wash was replaced with cell maintenance solution for continued culture. After 6 h, the cell culture medium was collected, adherent cells were released with digestion and the samples were centrifuged at 1000 rpm for 5 min. After washing the cell precipitates with PBS, the cells were resuspended by adding 500 μL of binding solution. Then, Annexin V-FITC and propidium iodide were added at 25 °C in the dark, and the mixture was mixed. Flow cytometry was performed after passing the samples through a 200-mesh filter.

### 2.6. Assay of the Antiviral Activity of Piceatannol against PRV In Vivo

Thirty female Kunming mice (20 ± 2 g) were purchased from Changsheng Laboratory Animal Center. After one week of acclimated feeding in the animal laboratory of Northeast Agricultural University, the mice were randomly divided into three groups: virus control group, the normal control group and piceatannol treatment group. The mice in the piceatannol treatment group were intraperitoneally injected with 50 mg/kg piceatannol once per day for 3 days, and the mice in the normal control group and the virus control group were injected with the same amount of DMEM. On day 4, the mice in the virus control group and the piceatannol treatment group were injected intramuscularly with 0.1 mL of 100 TCID_50_ PRV virus solution, and the mice in the normal control group were injected with the same amount of DMEM. Two hours after injection, the mice in the piceatannol administration group were intraperitoneally injected with piceatannol at a dose of 50 mg/kg once per day for 4 days, and the mice in the normal control group and the virus control group were injected with the same volume of DMEM. The health, behavior and body weight status of the mice were recorded daily. On the third day after challenge, three mice were randomly selected from each group for blood collection and dissection, and the heart, liver, lung, kidney and brain tissues of the mice were collected. At the same time, the remaining mice were monitored to determine the viability during the remaining experimental period. Tissue samples were homogenized in saline for DNA extraction to determine the tissue viral load. The other portion of specimens was fixed in 4% paraformaldehyde and then subjected to tissue section preparation and histopathological examination. The collected blood samples were coagulated at 4 °C for 20 min and then centrifuged at 5000 r/min for 10 min to obtain serum, and the concentrations of the TNF-α, IFN-γ, IL-6 and IL-4 cytokines in the mouse serum were measured using ELISA kits (Beijing Chenglin Biotechnology Co. Ltd., Beijing, China).

### 2.7. Statistical Methods

All cell experiments were repeated at least three times, and the results are presented as the mean ± standard deviation (SD). The statistical significance of the data was assessed with two-tailed Student’s *t* test using GraphPad Prism 8.0 software. *p* < 0.05 was considered to indicate statistical significance (ns: *p* > 0.05, * *p* < 0.05, ** *p* < 0.01, *** *p* < 0.001 and **** *p* < 0.0001).

## 3. Results

### 3.1. Cytotoxicity of Piceatannol on PK-15 Cells

The cytotoxicity of piceatannol on PK-15 cells was evaluated with CCK-8 assay. As shown in Figure 1, piceatannol inhibited the proliferation of PK-15 cells. With an increasing piceatannol concentration, the cell viability decreased gradually. When the concentration of piceatannol was less than 0.063 mg/mL, there was no significant difference in cell viability with the cell control group without treatment (*p* > 0.05). This concentration was selected as the nontoxic concentration for subsequent antiviral activity assays. The CC_50_ of piceatannol was 0.1129 mg/mL, as determined with nonlinear regression analysis.

### 3.2. Piceatannol Inhibited PRV Proliferation in PK-15 Cells

The inhibitory activity of piceatannol against PRV in the PK-15 cells was assessed by CCK-8 assay and FQ-PCR assay. The results showed that piceatannol significantly inhibited the cell death induced by the PRV infection in a dose-dependent manner. When the concentration of piceatannol was 0.063 mg/mL, the inhibition rate was as high as 82.78% (Figure 2B), the IC_50_ of piceatannol was 0.0307 mg/mL and the selectivity index (SI, CC_50_/IC_50_) was 3.68. In the viral copy number assay, the viral copy number decreased in a dose-dependent manner in the presence of piceatannol. When the concentration of piceatannol was 0.063 mg/mL, the viral copy number decreased by 13.57-fold (Figure 2C). In conclusion, piceatannol significantly inhibited PRV proliferation in PK-15 cells.

### 3.3. Effect of Piceatannol on the Replication Cycle of PRV

To further explore the effect of piceatannol on the PRV replication cycle, probe-based real-time PCR was used to determine the viral copy number in infected cells during the PRV adsorption, entry and replication phases. For adsorption assays, piceatannol and PRV were added simultaneously to the cells at 4 °C. Piceatannol significantly reduced the viral copy number of PRV and inhibited its adsorption (*p* < 0.0001) (Figure 3). For entry assays, cells were infected with PRV at 4 °C followed by the addition of a maintenance medium containing piceatannol. The results showed that there was no significant difference in viral load between the treatment group and the virus control group (*p* > 0.05) (Figure 3). This result indicates that piceatannol has no significant effect on virus entry. After entering the cell, the virus begins to replicate using the energy and raw materials provided by the cell. In the replication assay, piceatannol at a concentration of 0.063 mg/mL significantly reduced the viral copy number of infected cells and the replication of PRV (*p* < 0.05) (Figure 3).

### 3.4. Inhibitory Effect of Piceatannol on PRV Gene Expression

The expression levels of the PRV immediate early gene IE180, early gene EP0 and the PRV infection-related genes UL6, US27, UL44 and UL29 were measured with real-time PCR at 3, 6, 12, 24 and 48 h after virus infection as described previously. The relative expression levels of the detected genes are shown in Figure 4. The expression levels of the tested genes in the infected cells treated with piceatannol showed an increasing trend within 48 h. That is, with increasing time, the expression of the assayed genes increased. However, compared with that in the virus control group, the expression of all tested genes was significantly inhibited by piceatannol within 48 h (*p* < 0.05).

### 3.5. Piceatannol Alleviates PRV-Induced Apoptosis

The apoptosis of PK-15 cells was assessed with flow cytometry. The results showed that 36 h after viral infection, the apoptosis rate of normal control cells was 7.2% (Figure 5A), and that of PRV-infected control cells was 14% (Figure 5B). The apoptosis rate was 9.5% in the 0.063 mg/mL treatment group (Figure 5C). The results showed that PRV infection could induce the apoptosis of PK-15 cells, while piceatannol could reduce an apoptosis induced by PRV infection.

### 3.6. Piceatannol Inhibits PRV Infection in Mice

The mice in the control group began to develop neurological symptoms on the third day after infection, showing abnormal activity, biting at the injection site, having skin damage at the injection site and bleeding. Mice died on the third day after PRV challenge, and all the mice died by the fifth day, with a survival rate of 0%. The mice in the piceatannol group began to show clinical symptoms on the third day after challenge, and more died on the fourth day; however, one mouse survived to the seventh day, with a survival rate of 14.3%. All the mice in the nonchallenged normal control group survived (Figure 6A). The weight change curve of mice showed that compared with those in the normal control group, the mice in the virus control group continued to lose weight after the symptoms appeared on the third day. The mean body weight of the mice treated with piceatannol decreased at days four and five but increased significantly after day six (Figure 6B).

To determine the viral load in tissues and organs, three mice were randomly selected from each group, and the brain, liver, kidney, lung, and heart tissues were collected and weighed. Then, total DNA was extracted from tissues and organs, and changes in the viral DNA copy number were determined with FQ-PCR. The results showed that PRV was detectable in the brain, liver, kidney, lung and heart tissues of mice in the virus control group, and the virus loading in the brain tissue was the highest, followed by that in the liver, lung, kidney and heart. Compared with that of mice in the viral control group, the viral load in the brain and kidney was significantly reduced in the mice treated with piceatannol (*p* < 0.05), but there was no significant change in the virus loading in the heart, liver, and lung tissues (Figure 6C).

In order to further explore the protective effect of piceatannol on tissue injury caused by PRV, histopathological examination of the brain, heart, liver, lung and kidney of mice in each group was performed. The results of HE staining are shown in Figure 6D. Compared with those in the normal control group, the mice in the challenge group showed cellular edema and inflammatory cell infiltration in brain tissue. Rupture, rearrangement and a slight hemorrhage of myocardial fibers were observed in the heart. The liver showed extensive hepatocyte edema, lymphocyte infiltration and narrowing of the hepatic sinusoidal space. Pulmonary alveolar wall septal thickening and inflammatory cell infiltration were also observed. The renal tubular epithelial cells were swollen and congested, and the renal interstitium showed hemorrhage. In the mice treated with piceatannol, local lymphocyte infiltration in the brain tissue, myocardial hemorrhage in the heart, inflammatory cell infiltration in the liver and hepatocyte edema were reduced. The lung alveolar septum was thickened, and no inflammatory cell infiltration was observed. Swelling of renal epithelial cells was reduced, and a small amount of inflammatory cell infiltration was observed.

### 3.7. Changes in Cytokine Levels in the Serum of PRV-Infected Mice Treated with Piceatannol

Cytokines are involved in inflammatory and immune responses and play a key role in protecting the mechanism from foreign pathogens. The serum levels of IL-6, IFN-γ, TNF-α and IL-4 were detected with ELISA. The results are shown in Figure 7 Compared with that in the normal control mice, the PRV infection caused significant increases in the serum levels of IL-6, IFN-γ and TNF-α, (*p* < 0.05). However, the level of the cytokine IL-4 was significantly decreased. Compared with those in the mice in the challenge control group, the levels of TNF-α, IFN-γ and IL-4 cytokines in the serum of the mice in the piceatannol treatment group were significantly increased (*p* < 0.05). There was no significant difference in serum IL-6 levels between the challenge control group and the piceatannol treatment group (*p* > 0.05). The cytokine analysis results showed that a PRV infection could increase the levels of TNF-α, IFN-γ and IL-6 and reduce the level of IL-4 to regulate the body’s immune response to a virus infection.

## 4. Discussion

PRV is a widespread swine pathogen that causes severe neurological symptoms in infected pigs, reproductive disorders such as abortion and stillbirth and, ultimately, death in pregnant sows [14]. Once infected with the virus, it is difficult to eliminate the virus from the pig population, which has brought tremendous economic losses to the global pig industry. During this period, PRV vaccines were also developed, but due to the continuous mutation of PRV strains, traditional vaccines cannot provide comprehensive protection, and the disease is still spreading in China [15]. The potential for a threat to humans is also increasing [16]. Thus, new control measures are urgently needed.

Edible plant-derived compounds have multipathway and multitarget antiviral activities [17]. On the one hand, they can directly inhibit virus activity or replication processes, including blocking virus adsorption, entry, synthesis, mRNA replication or protein synthesis; on the other hand, they can regulate the body’s immunity to exert an antiviral activity [18]. Piceatannol is a naturally occurring polyphenolic substance in plants that is widely found in grapes, mushrooms, blueberries, passion fruit and other edible fruits [19]. Studies have reported that polyphenolic compounds have a wide range of antiviral activities [20]. For example, piceatannol has an inhibitory effect on human cytomegalovirus in vitro [12]. Epigallocatechin-3-gallate (EGCG), a bioactive polyphenol found in green tea extract, can inhibit PRV infection by inhibiting viral adsorption, entry and replication [21]. EGCG also inhibited PRRSV proliferation by interfering with lipid metabolism [22]. Moreover, EGCG showed inhibitory effects against influenza A virus, herpes simplex virus and Zika virus [23,24,25]. Curcumin is an acidic polyphenolic compound isolated from Curcuma longa. Yang et al. [26] found that curcumin could protect rat hippocampal neurons against a PRV infection by adjusting the BDNF/TrkB pathway. Studies have shown that curcumin can also inhibit the replication of CSFV by interfering with the lipid metabolism [27]. In addition, curcumin has an inhibitory effect on infectious gastroenteritis virus, hepatitis B virus and dengue 2 virus [28,29,30]. The above findings indicate that plant-derived polyphenolic compounds have a wide range of antiviral effects.

CCK-8 assay and CPE assay were used to determine the cytotoxicity of piceatannol and the inhibition rate of PRV by piceatannol. The antiviral activity of natural compounds is mainly demonstrated by their ability to reduce viral titers or inhibit CPE [31]. FQ-PCR is a method used for the quantification of viral DNA. The FQ-PCR showed that piceatannol could significantly reduce the copy number of PRV. The CCK8 assay showed that the activity of PK15 cells was significantly inhibited when the concentration of piceatannol was greater than 0.063 mg/mL. Moreover, the killing rate of piceatannol was as high as 82.78%. We then investigated the effect of piceatannol on the replication cycle of PRV. It was found that piceatannol inhibited the adsorption and replication of PRV. This mechanism is similar to the way panax notoginseng polysaccharide exerts its antiviral activity by blocking the PRV attachment to the cell surface and inhibiting PRV replication [32].

In the early stage of a PRV infection, PRV genes are transcribed, and some early viral proteins are expressed to further promote virus propagation [33]. Early genes are essential for progeny virus production after a PRV infection [34]. The expression levels of the PRV’s immediate early gene IE180 and early gene EP0 as well as the PRV infection-related genes UL6, US27, UL44 and UL29 were measured with RT-qPCR. The results showed that piceatannol could reduce the expression level of viral genes. This is consistent with the findings of Men et al. [35].

Innate immunity is an animal’s first line of defense against viral invasion. Apoptosis, as a cellular defense mechanism, plays a significant role in preventing virus diffusion in the early stage of virus infection, while it enhances virus replication and release in the late stage of virus infection [36]. Viral infection usually leads to apoptosis and inflammation [37]. Therefore, many viruses promote the production of antiapoptotic factors in the early period of infection to delay the death of cells, increase the production of virus and promote the release of more progeny viruses in the late stage of infection [38]. Studies have shown that both curcumin and myricetin can effectively inhibit viral infection by blocking PRV-induced apoptosis [26,39]. In this study, flow cytometry was used to examine the effect of piceatannol on host cell apoptosis at the late stage of PRV infection. The results confirmed that PRV infection could induce apoptosis of host cells, and piceatannol could inhibit PRV-induced apoptosis by inhibiting apoptosis, thereby limiting the production of progeny virions and inhibiting virus proliferation.

Studies have shown good antiviral activity of piceatannol in vitro. Therefore, we carried out an in vivo experimental study. Mice artificially infected with PRV were used as the animal model. The PRV inhibitory activity of piceatannol in mice was evaluated with an administration before and after challenge. The experiment showed that piceatannol had a protective effect on PRV-infected mice. Specifically, compared with those in the untreated virus control group, piceatannol administration delayed the onset of clinical symptoms, prolonged the average survival time and increased the viability of infected mice. These effects may be due to drug slowing the rate of virus proliferation in mice. Viral load is an important direct parameter used to evaluate antiviral effects in vivo [40]. It can reflect the replication of the virus in different organs [41]. In this study, the highest viral load was found in the brains of the virus control mice, which may be related to the neurotropic properties of alphaherpesviruses, which can invade the central nervous system through the trigeminal and sympathetic nerves [42]. Treatment with piceatannol significantly reduced the viral load in the brain and kidney tissues of the infected mice. Piceatannol reduces renal epithelial cell injury and has a neuroprotective effect [43]. We hypothesized that piceatannol could inhibit the replication of PRV in mouse organs. Meanwhile, the antiviral effect of piceatannol on PRV-infected mice was also confirmed with histopathological observation.

The inflammatory response is the first line of defense against the spread of viral infections. However, ensuring that the inflammation is resolved is necessary to prevent damage to the body. When the body’s inflammation is not resolved and not controlled, this condition will usually cause greater damage to the body [44]. After a viral infection, the body regulates the expression of cytokines such as interferons, tumor necrosis factor and interleukins through nonspecific immunity to protect cells from the viral infection. IFN-γ can activate macrophages; promote the production of the inflammatory factors TNF-α, IL-1β, and IL-12; and regulate the body’s immunity [45]. TNF-α, IFN-γ and IL-6, as important proinflammatory cytokines, can help the body fight a viral infection and prevent tissue damage by regulating the inflammatory response [46,47]. Therefore, in the present study, the serum levels of IFN-γ, IL-6 and TNF-α in mice were increased after challenge, indicating that the innate immunity of the mice was activated by PRV. The treatment with piceatannol significantly increased the serum levels of IFN-γ and TNF-α in the infected mice, suggesting that piceatannol could enhance the inflammatory response and inhibit virus replication at the early stage of virus infection by increasing the levels of TNF-α and IFN-γ. As an anti-inflammatory cytokine, IL-4 is mainly produced by Th2 cells. In the immune system, IL-4 not only regulates the immune function of macrophages but also plays a significant role in promoting the development of Th2 cells and inhibiting the growth of Th1 cells. In this study, the IL-4 cytokine level in the serum of infected mice was decreased, and it was significantly increased after the treatment with piceatannol. We hypothesized that piceatannol could modulate the Th1/Th2 balance by regulating IL-4 cytokine levels and then inhibit PRV infection. This is similar to how quercetin regulates Th1/Th2 balance by regulating IL-4 cytokine levels [48].

## 5. Conclusions

Piceatannol showed significant anti-PRV activity both in vitro and in vivo. When the concentration of piceatannol was 0.063 mg/mL, the inhibition rate was as high as 82.78%. Piceatannol could exert an anti-PRV effect by reducing the transcription level of viral genes, reducing PRV-induced apoptosis and elevating the levels of IL-4, TNF-α and IFN-γ in the serum of mice. Animal experiments showed that piceatannol could delay the onset of disease, reduce the viral load in the brain and kidney and reduce the pathological changes in the tissues and organs of the mice to improve the survival rate of the mice (14.3%). In summary, piceatannol has a protective effect against PRV infection in vivo and in vitro, which provides some new ideas for the treatment, prevention and control of PRV infection.

## Figures and Tables

**Figure 1 animals-13-02376-f001:**
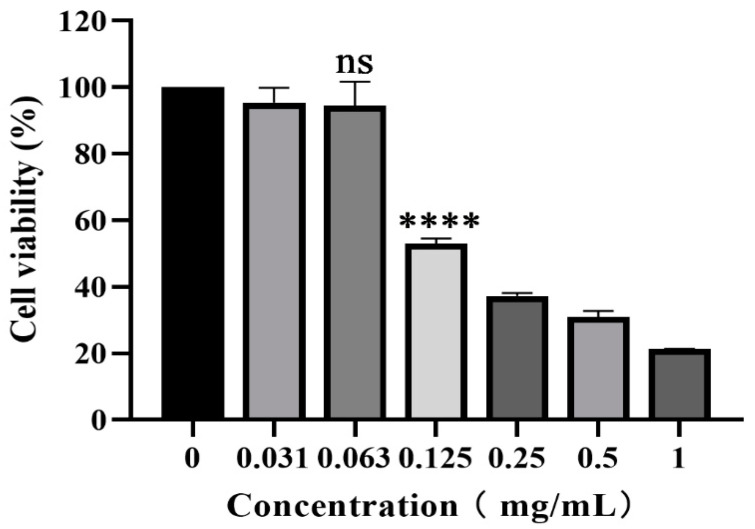
CCK-8 assay was used to determine the toxicity of piceatannol to PK-15 cells at 0.031, 0.063, 0.125, 0.25, 0.5 and 1 mg/mL (ns *p* > 0.05, **** *p* < 0.0001).

**Figure 2 animals-13-02376-f002:**
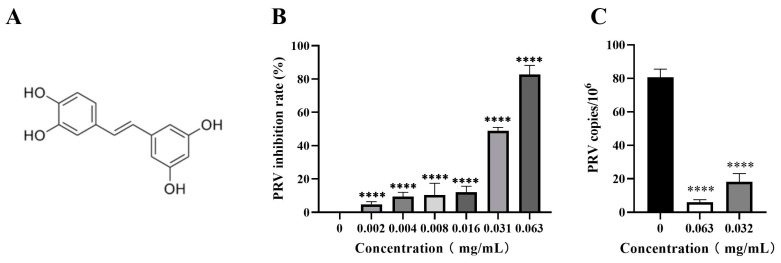
Piceatannol inhibited PRV proliferation in PK-15 cells. (**A**) Chemical structure of piceatannol. (**B**) CCK8 assay was used to detect the viability of PK-15 cells to calculate the rate of PRV inhibition by piceatannol. (**C**) FQ-PCR was used to measure the viral DNA copy number in PRV-infected cells treated with piceatannol (**** *p* < 0.0001).

**Figure 3 animals-13-02376-f003:**
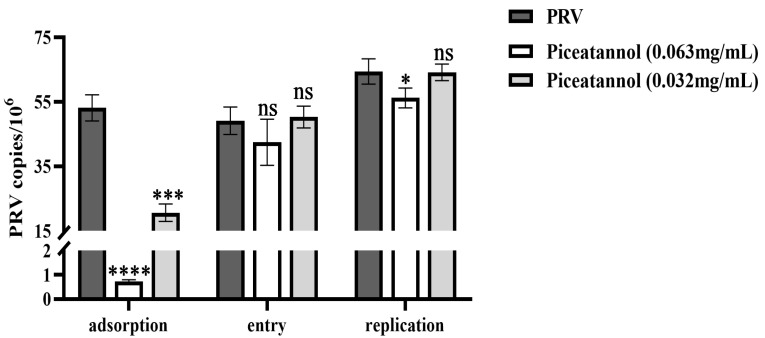
Piceatannol inhibited the PRV adsorption and replication phases (ns *p* > 0.05, * *p* < 0.05, *** *p* < 0.001 and **** *p* < 0.0001).

**Figure 4 animals-13-02376-f004:**
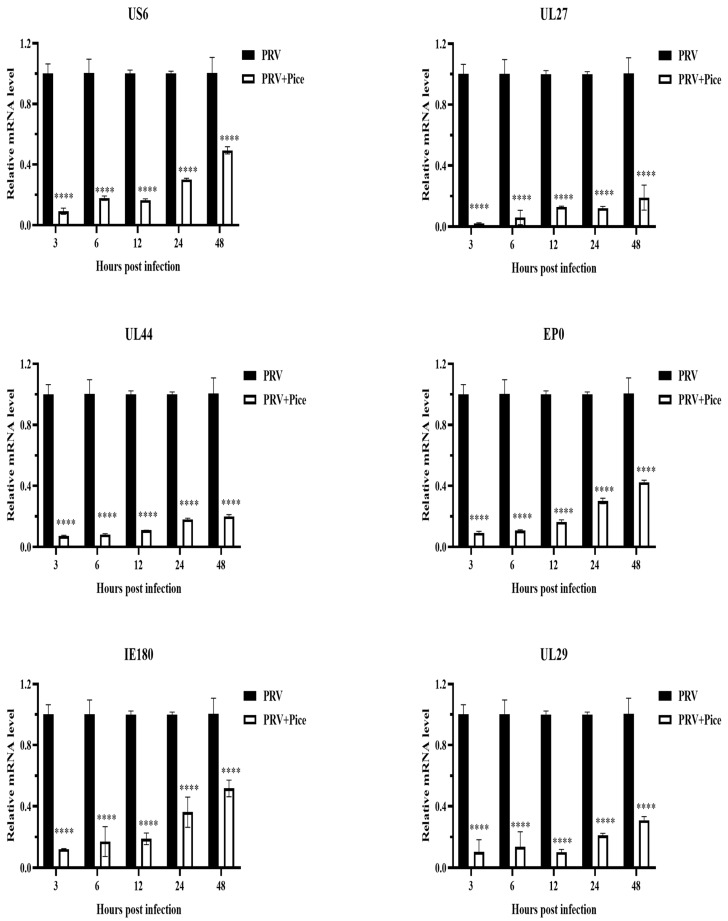
Piceatannol-inhibited PRV gene expression. Gene expression levels of PRV in the presence or absence of piceatannol (0.063 mg/mL) at 3, 6, 12, 24 and 48 hpi (**** *p* < 0.0001).

**Figure 5 animals-13-02376-f005:**
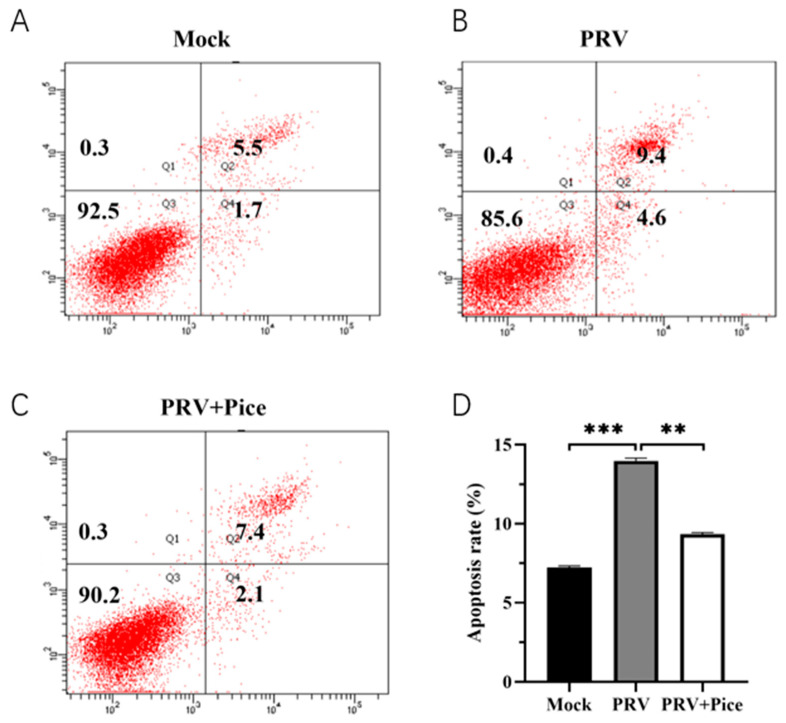
Piceatannol-attenuated PRV infection-induced apoptosis. After treatment with piceatannol (0.0625 mg/mL) for 36 h, the cells were stained with Annexin-V and PI and analyzed with flow cytometry. (**A**) Apoptosis rate in the cell control group; (**B**) apoptosis rate in the virus control group; (**C**) cell apoptosis rate of the piceatannol experimental group; (**D**) bar chart of cell apoptosis in each group (*** *p* < 0.001, ** *p* < 0.01).

**Figure 6 animals-13-02376-f006:**
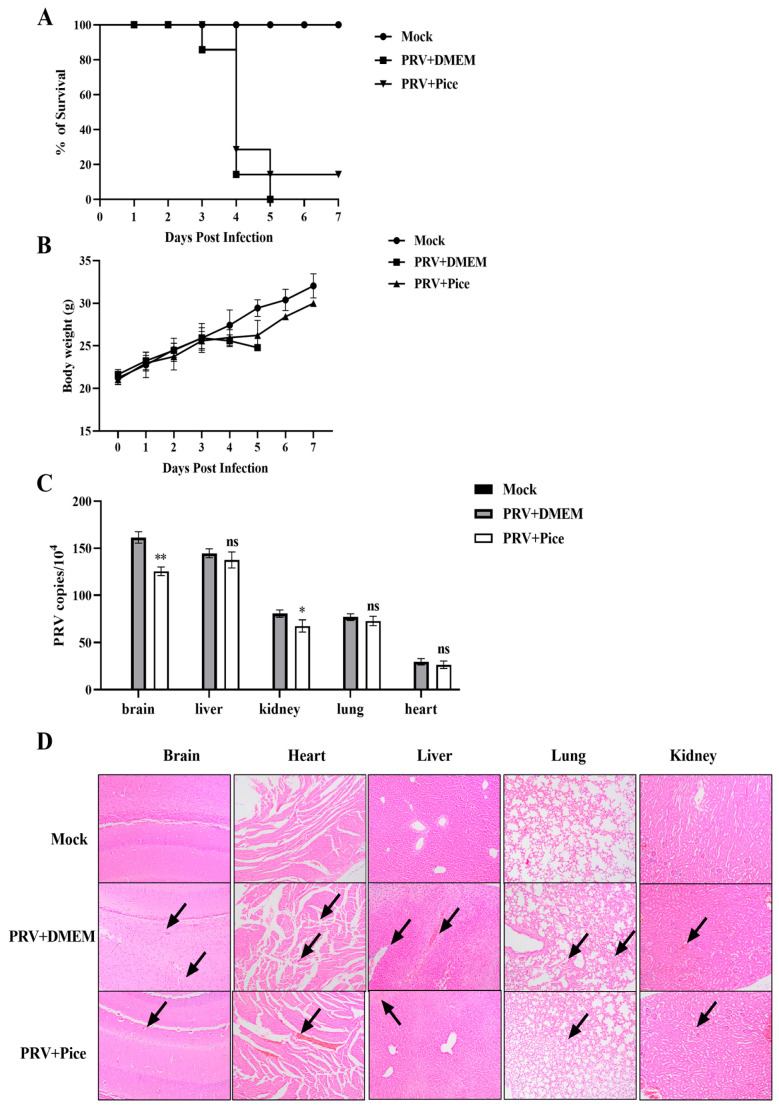
Piceatannol inhibits PRV activity in vivo. (**A**) Survival rate of mice in each group. (**B**) Changes in the body weight of mice in each group. (**C**) Viral load in tissues and organs of mice in each group. The heart, liver, lung, kidney and brain tissues of mice were collected on the third day after PRV challenge; viral DNA was extracted; and the viral copy number in each organ tissue was measured with FQ-PCR (ns *p* > 0.05, * *p* < 0.05, ** *p* < 0.01). (**D**) Pathological sections of mouse brain, heart, liver, lung and kidney tissues. The site shown by the black arrow is the lesion site.

**Figure 7 animals-13-02376-f007:**
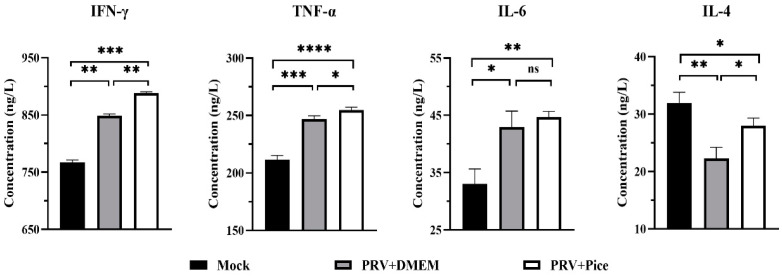
Serum levels of IFN-γ, TNF-α, IL-6 and IL-4 in each group. On day three after PRV infection, three mice from each group were randomly selected for blood collection, and the samples were used to quantify the four cytokines in mouse serum (ns *p* > 0.05, * *p* < 0.05, ** *p* < 0.01, *** *p* < 0.001, **** *p* < 0.001).

**Table 1 animals-13-02376-t001:** Primer sequences for real-time PCR.

Gene Name	Sequence (5′-3′)
UL44-F	CGTCAGGAATCGCATCA
UL44-R	CGCGTCACGTTCACCAC
IE180-F	CGCTCCACCAACAACC
IE180-R	TCGTCCTCGTCCCAGA
UL29-F	AGAAGCCGCACGCCATCACC
UL29-R	GGGAACCCGCAGACGGACAA
EP0-F	GGGCGTGGGTGTTT
EP0-R	GCTTTATGGGCAGGT
US6-F	AACATCCTCACCGACTTCA
US6-R	CGTCAGGAATCGCATCA
UL27-F	TCGTCCACGTCGTCCTCTTCG
UL27-R	CGGCATCGCCAACTTCTTCC
β-actin-F	TGCGGGACATCAAGGAGAA
β-actin-R	AGGAAGGAGGGCTGGAAGA

## Data Availability

The original contributions presented in the study are included in the article, and further inquiries can be directed to the corresponding author(s).

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
