# Peer review of "Piceatannol as an Antiviral Inhibitor of PRV Infection In Vitro and In Vivo"

_animals, 2023, doi:10.3390/ani13142376_

Round 1

Reviewer 1 Report

The work described by Wang et al. describes the antiviral activity of Piceatannol in vitro and in vivo on Pseudorabies infection. The work is of undisputed scientific interest, although it is not written in an appropriate manner and presents numerous experimental flaws, which are described below. In my opinion, the manuscript requires a major overhaul both methodologically and with regards to its structure and the quality of the English. For these reasons, the work is not yet publishable.

General concept comments:

Formatting problems: The number line is missing. A simple summary is missing. "In vivo" and "in vitro" are not suitable keywords. English is very poor (Some sentences are too simplistic, others are very convoluted). The introduction requires a more in-depth study of the PRV (it seems that the attention of the authors is focused almost exclusively on the Piceatannol). The materials and methods section should be simpler and easier to interpret. There are also some methodological weaknesses. For example, the part "Effect of piceatannol on the PRV life cycle" is not clear. Some parts of the materials and methods are repetitive. The point of evaluating the expression of different viral proteins is not understood. What is the purpose of these experiments carried out by the authors? The same is true for western blot analysis. Why were two different proteins evaluated? The quality of the western blot image (ie of the whole blot) is insufficient; the blot is cut, and it is not possible to understand the molecular weight or whether the published actin is actually that of the membrane placed in the supplementary file. PRV is a virus that induces apoptosis in its permissive cells (PK-13). The reduction of protein and nucleic acid expression does not correspond to a drastic reduction of early and late apoptosis events. Authors should explain this and try to explain it. The standard deviation of the graph relating to the analysis with the FACS seems very low to me. How many times have these experiments been carried out? In the discussion section, the authors report the results obtained without adequate comparison with other similar studies using other natural substances or antiviral drugs. Nonetheless, 72 articles were cited, far too many (67 would be more than fine). Another weakness of the article consists of some statements in which it seems that the authors want to promote the use of a therapy based on this natural product to be able to treat animals suffering from pseudorabies. Surely the effects that these substances have on the viral replicative cycle are interesting, but there is not enough scientific evidence to be able to make certain types of statements.

Specific comments:

Title: "Prv" should be "PRV" or "Pseudorabies virus"

Abstract:

The abstract is long and very brief on the introduction, the methods used, the results obtained, and the implications of this study.

Introduction:

A more comprehensive and recent literature review is required, especially to provide more information about PRV, its impact, spread, and epidemiology. I recommend this work to enrich the introduction to the presence of PRV among wildlife: doi: 10.3390/ani11113298.

Materials and methods:

Materials and Methods has numerous inaccuracies already described as general comments. Furthermore, the origin of the compound is not entirely clear. Is the PCR used described in the literature or were the primers designed by the authors? Wouldn't it be better to talk about ul/ml? PCR extraction and setup kits lack manufacturer information.

The authors wrote, "the 3D structure of piceatannol was downloaded from PubChem, and the 3D structures of the PRV gB (PDB ID: 5YS6) protein and PRV gD (PDB ID: 5X5V) protein were downloaded from the RCSB PDB online database. PyMOL software was used to optimize the protein structure and remove water and small molecular ligands. AutoDock was used to hydrogenate the PRV gB/gD protein molecular model. GridBox was used to wrap the whole protein molecules for molecular docking. The optimal docking results with the lowest binding energy were screened and visualized by PyMOL." In the subsection "Effect of piceatannol on PRV protein expression". I don't understand why this evaluation was made or why it is included in this subsection.

The methods of infection (often always the same) are repeated continuously throughout the manuscript.

Results:

The results seem incomplete. Inhibition data for the least effective doses are lacking. The results relating to adsorption, etc., have no scientific significance on my part and should be discarded.

I recommend the authors thoroughly revise the manuscript and greatly improve their English.

The manuscript requires English editing. It should be done after further reviewing the manuscript and the article changes from major revision to minor / accepted.

Author Response

Response to Reviewer 1Comments

Dear Editors and Reviewer,

Thank you for your work and comment. The comments and suggestions are very helpful for revising and improving our paper and research. We have studied every comment carefully and made corrections one by one. For your valuable comments, words in red are the revisions we have made in the manuscript. Generally, this study made the following responses:

Point 1: The number line is missing.

Response 1: Thank you very much and we are very sorry for our mistake. I have added the line number.

Point 2: English is very poor (Some sentences are too simplistic, others are very convoluted)

Response 1: We appreciate the reviewers’ suggestion. To improve the grammar and readability, the newly submitted paper has been carefully revised by a professional language editing service.

Point 3: "In vivo" and "in vitro" are not suitable keywords.

Response 3: Thank you very much for your valuable advice. We have deleted “In vivo” and “In vitro” in line 23.

Point 4: The materials and methods section should be simpler and easier to interpret. There are also some methodological weaknesses. For example, the part "Effect of piceatannol on the PRV life cycle" is not clear.

Response 4: Thank you very much for your comments. I have made some modifications to the test method of “ Effect of piceatannol on the PRV life cycle ” again in line 76-87.

Point 5: The point of evaluating the expression of different viral proteins is not understood. What is the purpose of these experiments carried out by the authors?

Response 5: Thank you very much for your questions. We detect viral proteins for two purposes. The first is that these two proteins are important functional proteins of PRV. Second, we detected the antiviral effect of piceatannol at the gene level by fluorescent quantitative PCR. So we detect the viral protein and once again verify the antiviral effect of the piceatannol.

Point 6: The quality of the western blot image (ie of the whole blot) is insufficient;  the blot is cut, and it is not possible to understand the molecular weight or whether the published actin is actually that of the membrane placed in the supplementary file.

Response 6: We appreciate this commet very much. We have resubmitted the gB and gD protein original gel pictures to the editor.

Point 7: PRV is a virus that induces apoptosis in its permissive cells (PK-13). The reduction of protein and nucleic acid expression does not correspond to a drastic reduction of early and late apoptosis events. Authors should explain this and try to explain it.

Response 7: Thank you very much for your questions. The antiviral effects of monomeric compounds are multi-target and multi-pathway. Piceatannol can inhibit viral replication by inhibiting viral proteins to exert antiviral activity. At the same time, it can also exert antiviral activity by inhibiting viral gene transcription level and inhibiting virus-induced apoptosis. The protein expression test and nucleic acid expression test were not necessarily related to the apoptosis test.

Point 8: The standard deviation of the graph relating to the analysis with the FACS seems very low to me. How many times have these experiments been carried out?

Response 8: Thank you very much for your questions. We conducted three experiments on FACS, and the results of all three experiments were basically consistent. Although the differences in FACS results are not significant. However, this result can reflect that piceatannol can reduce PRV-induced apoptosis.

Point 9: In the discussion section, the authors report the results obtained without adequate comparison with other similar studies using other natural substances or antiviral drugs.

Response 9: Thank you very much for your valuable suggestions. We made some changes to remove inappropriate parts of the discussion.

Point 10: Nonetheless, 72 articles were cited, far too many (67 would be more than fine).

Response 10: Thank you very much for your questions. I have reduced the number of references in the paper to 50.

Point 11: Surely the effects that these substances have on the viral replicative cycle are interesting, but there is not enough scientific evidence to be able to make certain types of statements.

Response 11: Thank you very much for your questions. The virus replication cycle test was conducted according to the following literature. Viral replication is a very complex process, and the complete replication mechanism of PRV has not been fully revealed. What we did was just a preliminary study. So far we've only found that drugs work in that replication phase. The next step will be to conduct more in-depth research on this aspect.

[1] Cai X, Shao Y, Wang Z, Xu Y, Ren Z, Fu L, Zhu Y. Antiviral activity of dandelion aqueous extract against pseudorabies virus both in vitro and in vivo. Front Vet Sci. 2023 Jan 9;9:1090398. doi: 10.3389/fvets.2022.1090398. PMID: 36699332; PMCID: PMC9870063.

[2] Men X, Li S, Cai X, Fu L, Shao Y, Zhu Y. Antiviral Activity of Luteolin against Pseudorabies Virus In Vitro and In Vivo. Animals (Basel). 2023 Feb 20;13(4):761. doi: 10.3390/ani13040761. PMID: 36830548; PMCID: PMC9952634.

[3] Li L, Wang R, Hu H, Chen X, Yin Z, Liang X, He C, Yin L, Ye G, Zou Y, Yue G, Tang H, Jia R, Song X. The antiviral activity of kaempferol against pseudorabies virus in mice. BMC Vet Res. 2021 Jul 18;17(1):247. doi: 10.1186/s12917-021-02953-3. PMID: 34275451; PMCID: PMC8287772.

[4] Zhao X, Chen Y, Zhang W, Zhang H, Hu Y, Yang F, Zhang Y, Song X. Dihydromyricetin Inhibits Pseudorabies Virus Multiplication In Vitro by Regulating NF-κB Signaling Pathway and Apoptosis. Vet Sci. 2023 Feb 2;10(2):111. doi: 10.3390/vetsci10020111. PMID: 36851415; PMCID: PMC9961748.

[5] Huan C, Zhou Z, Yao J, Ni B, Gao S. The Antiviral Effect of Panax Notoginseng Polysaccharides by Inhibiting PRV Adsorption and Replication In Vitro. Molecules. 2022 Feb 13;27(4):1254. doi: 10.3390/molecules27041254. PMID: 35209042; PMCID: PMC8880127.

[6] Huan C, Xu W, Guo T, Pan H, Zou H, Jiang L, Li C, Gao S. (-)-Epigallocatechin-3-Gallate Inhibits the Life Cycle of Pseudorabies Virus In Vitro and Protects Mice Against Fatal Infection. Front Cell Infect Microbiol. 2021 Jan 14;10:616895. doi: 10.3389/fcimb.2020.616895. PMID: 33520741; PMCID: PMC7841300.

Point 12: "Prv" should be "PRV" or "Pseudorabies virus".

Response 12: We appreciate this commet very much and we have modified Prv to PRV in line 1.

Point 13: The abstract is long and very brief on the introduction, the methods used, the results obtained, and the implications of this study.

Response 13: Thank you very much for your valuable suggestions. I have made a brief revision of the summary in line 14-22.

Point 14: A more comprehensive and recent literature review is required, especially to provide more information about PRV, its impact, spread, and epidemiology. I recommend this work to enrich the introduction to the presence of PRV among wildlife: doi: 10.3390/ani11113298.

Response 14: Thank you very much for your valuable advice. I carefully read the excellent article entitled " Aujeszky’s Disease in South-Italian Wild Boars (Sus scrofa): A Serological Survey

" and quoted it in [1] of this article. Since the literature of [2] is quite old, I have replaced it with a new one in line 29 and 32.

Point 15: Materials and Methods has numerous inaccuracies already described as general comments.

Response 15: We appreciate this commet very much. I have corrected some language errors in the materials and methods in line 76-88..

Point 16: Furthermore, the origin of the compound is not entirely clear.

Response 16: We appreciate this commet very much. Piceatannol was purchased from Dalian Meilun Biotechnology Co., LTD.

Point 17: Is the PCR used described in the literature or were the primers designed by the authors?

Response 17: We appreciate this commet very much. All the primers in the manuscript were designed by us.

Point 18: PCR extraction and setup kits lack manufacturer information.

Response 18: We appreciate this commet very much. I've added the manufacturer in line 59, 89, 100, 126 and 161.

Point 19: The authors wrote, "the 3D structure of piceatannol was downloaded from PubChem, and the 3D structures of the PRV gB (PDB ID: 5YS6) protein and PRV gD (PDB ID: 5X5V) protein were downloaded from the RCSB PDB online database. PyMOL software was used to optimize the protein structure and remove water and small molecular ligands. AutoDock was used to hydrogenate the PRV gB/gD protein molecular model. GridBox was used to wrap the whole protein molecules for molecular docking. The optimal docking results with the lowest binding energy were screened and visualized by PyMOL." In the subsection "Effect of piceatannol on PRV protein expression". I don't understand why this evaluation was made or why it is included in this subsection.

Response 19: We appreciate this commet very much. We first investigated the inhibitory effect of piceatannol on PRV replication cycle by affecting PRV gB and gD proteins. We then used molecular docking technology to explore the specific binding sites of piceatannol to viral proteins. I changed the subtitle in line 107.

Point 20: The methods of infection (often always the same) are repeated continuously throughout the manuscript.

Response 20: We appreciate this commet very much. I modified the repeated infection methods in line 88, 97-98 and 115-117.

Point 21: The results seem incomplete. Inhibition data for the least effective doses are lacking. The results relating to adsorption, etc., have no scientific significance on my part and should be discarded.

Response 21: Thank you very much for your valuable advice. I added the inhibition rate of piceatannol in line 440-441. The replication cycle of PRV is divided into viral adsorption, entry, uncoating, viral genome transcription and replication, viral protein synthesis, and progeny virus release. Each stage of the replication cycle is meaningful.

We try our best to improve the manuscript and have made some changes marked in red in revised manuscript which will not influence the content and framework. We appreciate Editors and Reviewer’ warm work earnestly, and hope the revision will meet with approval. Once again, thank you very much for your comments and suggestions.

Kind regards, 
Yan Zhu
Corresponding author

Reviewer 2 Report

The manuscript describes the compound Piceatannol and supports evidences for it being a potent antiviral against PRV. In addition to the well known anti-inflammatory, anticancer, antiaging properties of the compound, the manuscript provides evidence for its antiviral characteristics. There are a few minor changes that I have suggested below:

The PRV gene expression levels were measured using RT-qPCR. The PRV-gB and gD proteins were also measured with immunoblotting. However, the downstream effects of piceatannol on the immune signaling proteins of cells infected with PRV has not been demonstrated.

The in-vivo work has been successfully demonstrated. Serum levels of certain inflammatory factors have been shown to be regulated upon PRV infection and piceatannol treatment. However, I believe more evidence could be provided to support the hypothesis that piceatannol modulates the Th1/Th2 balance.

Similar to the evidence provided for the immune gene regulation in serum levels of mice, it would be a great addition to describe some trends in the cell lines.

All figure legends could be modified to briefly include the results in addition to mentioning information about the procedure involved.

The introduction section, in my opinion contains a significant amount of material irrelevant to the work described in the manuscript. Rather more details could be included about the past work done on piceatannol or similar compounds with regards to their anti-PRV related properties. Although there is one sentence mentioned in the end of the introduction section about what has been done in this study, I believe it could be elaborated to make the section relevant to the study performed. 

Author Response

Response to Reviewer 2Comments

Dear Editors and Reviewer,

Thank you for your work and comment. The comments and suggestions are very helpful for revising and improving our paper and research. We have studied every comment carefully and made corrections one by one. For your valuable comments, words in red are the revisions we have made in the manuscript. Generally, this study made the following responses:

Point 1: The PRV gene expression levels were measured using RT-qPCR. The PRV-gB and gD proteins were also measured with immunoblotting. However, the downstream effects of piceatannol on the immune signaling proteins of cells infected with PRV has not been demonstrated.

Response 1: We appreciate this commet very much. We have only preliminarily explored the effects of Piceatannol on two important functional proteins of PRV. We will continue to explore the downstream effects of piceatannol on the immune signaling proteins of cells infected with PRV in future studies.

Point 2: The in-vivo work has been successfully demonstrated. Serum levels of certain inflammatory factors have been shown to be regulated upon PRV infection and piceatannol treatment. However, I believe more evidence could be provided to support the hypothesis that piceatannol modulates the Th1/Th2 balance.

Response 2: Thank you very much for your valuable suggestions. I have added relevant literature [50].

Point 3: Similar to the evidence provided for the immune gene regulation in serum levels of mice, it would be a great addition to describe some trends in the cell lines.

Response 3: Thank you very much for your valuable suggestions. I have supplemented some results at the cellular level in the discussion section

Point 4: All figure legends could be modified to briefly include the results in addition to mentioning information about the procedure involved.

Response 4: Thank you very much for your valuable suggestions. I have simplified the description of some of the legends in the results

Point 5: The introduction section, in my opinion contains a significant amount of material irrelevant to the work described in the manuscript. Rather more details could be included about the past work done on piceatannol or similar compounds with regards to their anti-PRV related properties. Although there is one sentence mentioned in the end of the introduction section about what has been done in this study, I believe it could be elaborated to make the section relevant to the study performed. 

Response 5: Thank you very much for your valuable suggestions. I have removed the second part of the introduction.

We try our best to improve the manuscript and have made some changes marked in red in revised manuscript which will not influence the content and framework. We appreciate Editors and Reviewer’ warm work earnestly, and hope the revision will meet with approval. 
Once again, thank you very much for your comments and suggestions.

Kind regards,
Yan Zhu
Corresponding author.

Reviewer 3 Report

The quality of English language in the manuscript is acceptable. However, there is room for improvement. Some sentences could be rephrased to enhance clarity and precision. Additionally, attention should be given to grammar and word choice to ensure accurate communication of ideas.

Author Response

Response to Reviewer 3Comments

Dear Editors and Reviewer,

Thank you for your work and comment. The comments and suggestions are very helpful for revising and improving our paper and research. We have studied every comment carefully and made corrections one by one. For your valuable comments, words in red are the revisions we have made in the manuscript. Generally, this study made the following responses:

Point 1: Regarding the study design, the authors conducted the detection of PRV-related genes and partial structural proteins in vitro but did not perform corresponding validation in vivo. Additionally, the in vivo experiments were conducted using mice instead ofpigs. It is recommended that the authors provide a reasonable explanation for the lack ofvalidation of genes and proteins in the in vivo experiments and the choice ofmice as the experimental subjects instead of pigs.

Response 1: We appreciate this commet very much. PRV has a broad host infection range and is capable of infecting most mammals. Mice, rabbits, etc. are standard test animals. We first investigated the anti-PRV activity of Piceatannol in vitro. Then in vivo experiments were performed in mice. Survival rate, body weight change, viral copy number in visceral tissue, histopathological changes and serum cytokines were detected. The data of in vivo experiments in mice are relatively sufficient.

Abstract:

Point 2: The repetition of yhe phrase,”the IC50 was 0.0307 mg/ml” in the abstract should be checked for accuracy and one of the repetitions should be removed.

Response 2: Thank you very much for your valuable advice. I have deleted “ the IC50 was 0.0307 mg/ml ” .

Introduction

Point 1: The first paragraph of the introduction describes the harm caused by PRV,the second

paragraph introduces the role of natural products, and the third paragraph introduces

the functions of Piceatannol. Overall, the design is reasonable. Since Piceatannol

belongs to a bioactive polyphenol substance, it is recommended to emphasize the

efficacy of bioactive polyphenol substances in the second paragraph to provide better

context and background information for the readers.

Response 1: Thank you very much for your valuable advice. I have deleted the irrelevant content in the second paragraph of the introduction as proposed by Reviewer 2. The effects of biopolyphenols are briefly described in the third paragraph of the introduction.

Point 2: It is suggested to provide a more detailed and accurate description of the research significance in order to avoid repetition with the abstract.

Response 2: Thank you very much for your valuable advice. I have removed the duplicate part of the abstract.

Point 3: Please verify the spelling of the word"linosides" in the sentence"The linosides and

alkaloids in peppers can inhibit hepatitis B virus pathogenesis [16]."

Response 3: Thank you very much and we are very sorry for our mistake. I have deleted the irrelevant content in the second paragraph of the introduction as proposed by Reviewer 2.

Materials and Methods

Point 1: The sentence "On the third day after challenge,three mice were randomly selected from each group for blood collection and dissection, and the heart,liver, lung, kidney

and brain tissues of the mice were collected " is inconsistent with the earlier statement

of" 50 mg/kg piceatannol once per day for 4 days."It is suggested that the authors

clarify whether the sampling time point is consistent with the treatment time point

during the experiment and provide a specific explanation for the sample collection on

the third day.

Response 1: We appreciate this commet very much. We treated the mouse as day 0 after PRV injection. The third day after challenge was actually four days. So there is no conflict “50 mg/kg once per day for 4 days” and “On the third day after challenge, three mice were randomly selected from each group for blood collection and dissection, and the heart, liver, lung, kidney and brain tissues of the mice were collected”.

Results

Point 1: The authors state that "The results showed that piceatannol significantly inhibited

cell death induced by PRV infection in a dose-dependent manner," but no relevant

results are presented in the text.Furthermore, there is no direct evidence to support the

claim that piceatannol can inhibit PRV-induced cell death. It is recommended that the

authors provide relevant results to support the claim of piceatannol's inhibition of PRV-

induced cell death and consider discussing the results of the CCK-8 experiment in the

text.

Response 1: We appreciate this commet very much. It can be clearly seen in Figure. 2B that the inhibition rate of PRV increased with the increase of piceatannol concentration. In Figure 2C, it can be found that the viral copy number decreased with the increase of piceatannol concentration.

Thank you very much and we are very sorry for our mistake. This expression is incorrect. I will change inhibit to reduce in line 259 and 443.

Thank you very much for your valuable advice. I have added the CCK8 experimental results in the discussion section in lin 362-363 and 365-368.

Point 2: Why there is no significiant difference labeled in Figure 2B,bar graph?

Response 2: Thank you very much and we are very sorry for our mistake. I have resubmitted Figure 2.

Point 3: Figure 3 is not labeled in the text.

Response 3: Thank you very much and we are very sorry for our mistake. I have marked Figure 3 in the text.

Point 4: The image quality of Figure 4 is poor and needs improvement.

Response 4: We appreciate this commet very much. I have re-uploaded Figure 4.

Point 5: In Figures 5A and 5C, the immunoblot images appe car to have different exposure levels, with noticeable differences in darkness. The au thors should strive to ensure consistency in the image quality.

Response 5: Thank you very much and we are very sorry for our mistake. Because the chemiluminescence instrument in our laboratory was damaged, Figure 5C was exposed using instruments from other laboratories. So there's a difference in exposure levels between these two images. However, we are trying very hard to adjust it. Please understand the inconvenience caused to you.

Point 6: In the text, it is mentioned that "The mice in the piceatannol group began to show clinical symptoms on the 3rd day after challenge and died on the 4th day, and one mouse survived to the 7th day, with a survival rate of 14.3%." However, if there were 10 mice in the group and only one survived until the seventh day,the survival rate should not be 14.3%. Please check and explain the accuracy of the survival rate calculation and ensure the accuracy of image labeling.

Response 6: We appreciate this commet very much. From the 10 mice, three mice were randomly selected for blood sampling and dissection, thus leaving seven mice with a final survival rate of one in seven (14.3%).

Point 7: In Figure 7D's histopathology image, please mark the specific location ofthe lesions with arrows or asterisks. Additionally, each image should have a scale bar.

Response 7: We appreciate this commet very much. I have uploaded Figure 7D again.

Point 8: In the earlier Methods section, it is mentioned that the samples were collected on the third day after challenge, and the mice in the virus group had mostly died by the second day after treatment. Therefore, were thescollected samples from already deceased mice? Can the authors confirm the consistency of the time of death for all samples? If there are inconsistencies, how are these addressed?Please explain the time points of sample collection in Figures 7 and 8 and provide an explanation in the Discussion section.

Response 8: We appreciate this commet very much. When the mice in the virus group began to die (on the third day after challenge), three mice were randomly selected from the surviving mice for blood sampling and dissection. Both Figures 7 and 8 samples were collected at time points 3 days after challenge in mice. We try our best to improve the manuscript and have made some changes marked in red in revised manuscript which will not influence the content and framework. We appreciate Editors and Reviewer’ warm work earnestly, and hope the revision will meet with approval. Once again, thank you very much for your comments and suggestions.

Kind regards,
Yan Zhu
Corresponding author

Round 2

Reviewer 1 Report

I am glad to note that the authors have responded to most of the revisions I have raised and have greatly improved the quality of the manuscript. Thanks to the editing of the English, the manuscript is definitely more flowing and easier to read. However, I find myself forced to request further major revisions from the authors, due to the western blot images. Such westerns do not meet the minimum image requirements estabilished in the journal's western blot analysis. In particular, the blots were cut in two (which is allowed), but the actin and the gD/gI should be on the same piece of membrane. The way the authors show the experiment, it is not possible to know whether those actins actually correspond to the actins of the membranes. Furthermore, even the cutting lines would appear to be non-superimposable. I advise the authors to show whole images of the blot before being cut; if not available, run these membranes again without cutting them (or, if not possible, delete this part of the data). Below are some minor comments.

Line 11: Please change "antiherpesvirus effect" to " antiviral activity"

Line 12: Please delete "In this study, we evaluated the antiviral activity of piceatannol against PRV" and continue the previous sentence.

Line 22: The authors wrote "be a novel antiherpesvirus infection agent in the future". This statement should be modified, as it would appear that the authors propose the natural substance they studied as an antiviral drug to be administered to PRV-infected animals. Although the compound they have studied has antiviral effects in vivo and in vitro, it is unlikely that it will ever be used, for example, to resolve a PRV outbreak (where direct measures and vaccination remain the most effective measures). I advise the authors to modify these sentences throughout the manuscript, discussing the auxiliary role that these substances can have.

Line 34-35: We agree with the authors that the latest research would have found the infection in humans, however, it occurs only under certain conditions. So PRV is far from being a risk to human health. Please delete lines 34-35.

Line 42: Ripetition, delete it. "It is important for human health, nutrition and well-being".

Line 61: Please, change "continuous" to "serial" and "dilution" to "dilutions".

Line 338: As Line 22. Please, resentence it.

The level of English has definitely improved after editing, and the manuscript now appears more understandable.

Author Response

Response to Reviewer 1Comments

Dear Editors and Reviewer,

Thank you for your work and comment. The comments and suggestions are very helpful for revising and improving our paper and research. We have studied every comment carefully and made corrections one by one. For your valuable comments, words in red are the revisions we have made in the manuscript. Generally, this study made the following responses:

Point 1: However, I find myself forced to request further major revisions from the authors, due to the western blot images. Such westerns do not meet the minimum image requirements estabilished in the journal's western blot analysis. In particular, the blots were cut in two (which is allowed), but the actin and the gD/gI should be on the same piece of membrane. The way the authors show the experiment, it is not possible to know whether those actins actually correspond to the actins of the membranes. Furthermore, even the cutting lines would appear to be non-superimposable. I advise the authors to show whole images of the blot before being cut; if not available, run these membranes again without cutting them (or, if not possible, delete this part of the data).

Response 1: Thank you very much for your valuable advice. I am very sorry that the WB test results do not meet the requirements of this journal. I intend to redo the WB experiment as requested by the journal. But the antibodies for the WB test were exhausted. So I had to delete the data about the WB trial.

Point 2: Line 11: Please change "antiherpesvirus effect" to " antiviral activity.

Response 2: Thank you very much for your valuable advice. I replaced antiherpesvirus effect with antiviral activity.

Point 3: Line 12: Please delete "In this study, we evaluated the antiviral activity of piceatannol against PRV" and continue the previous sentence”.

Response 3: Thank you very much for your valuable advice. I have deleted “In this study, we evaluated the antiviral activity of piceatannol against PRV" and continue the previous sentence”.

Point 4: The authors wrote "be a novel antiherpesvirus infection agent in the future".  This statement should be modified, as it would appear that the authors propose the natural substance they studied as an antiviral drug to be administered to PRV-infected animals.  Although the compound they have studied has antiviral effects in vivo and in vitro, it is unlikely that it will ever be used, for example, to resolve a PRV outbreak (where direct measures and vaccination remain the most effective measures).  I advise the authors to modify these sentences throughout the manuscript, discussing the auxiliary role that these substances can have.

Response 4: Thank you very much for your valuable advice. I replaced the " be a novel antiherpesvirus infection agent in the future " with another expression in line 19-21, 32-34, 58-59, 301 and 401-403.

Point 5: We agree with the authors that the latest research would have found the infection in humans, however, it occurs only under certain conditions. So PRV is far from being a risk to human health. Please delete lines 34-35.

Response 5: Thank you very much for your valuable advice. I've deleted it.

Point 6: Line 42: Ripetition, delete it. "It is important for human health, nutrition and well-being".

Response 6: Thank you very much for your valuable advice. I have deleted “It is important for human health, nutrition and well-being”.

Point 7: Line 61: Please, change "continuous" to "serial" and "dilution" to "dilutions".

Response 7: Thank you very much for your valuable advice. I've replaced "continuous" to "serial" and "dilution" to "dilutions".

Point 8: Line 338: As Line 22. Please, resentence it.

Response 8: Thank you very much for your valuable advice. I've rephrased it.

We try our best to improve the manuscript and have made some changes marked in red in revised manuscript which will not influence the content and framework. We appreciate Editors and Reviewer’ warm work earnestly, and hope the revision will meet with approval. Once again, thank you very much for your comments and suggestions.

 Kind regards, 

Yan Zhu

 Corresponding author.
